# Characterization of a Cell Culture System of Persistent Hepatitis E Virus Infection in the Human HepaRG Hepatic Cell Line

**DOI:** 10.3390/v13030406

**Published:** 2021-03-04

**Authors:** Marie Pellerin, Edouard Hirchaud, Yannick Blanchard, Nicole Pavio, Virginie Doceul

**Affiliations:** 1UMR 1161 Virologie, INRAE, ANSES, Ecole Nationale Vétérinaire d’Alfort, Université Paris-Est, 94700 Maisons-Alfort, France; marie.pellerin@anses.fr (M.P.); nicole.pavio@anses.fr (N.P.); 2Agence Nationale de Sécurité Sanitaire, De L’environnement et du Travail (ANSES), Laboratory of Ploufragan-Plouzané-Niort, Viral Genetic and Biosafety (GVB) Unit, 22440 Ploufragan, France; edouard.hirchaud@anses.fr (E.H.); yannick.blanchard@anses.fr (Y.B.)

**Keywords:** hepatitis E virus, cell culture model, persistent infection

## Abstract

Hepatitis E virus (HEV) is considered as an emerging global health problem. In most cases, hepatitis E is a self-limiting disease and the virus is cleared spontaneously without the need of antiviral therapy. However, immunocompromised individuals can develop chronic infection and liver fibrosis that can progress rapidly to cirrhosis and liver failure. The lack of efficient and relevant cell culture system and animal models has limited our understanding of the biology of HEV and the development of effective drugs for chronic cases. In the present study, we developed a model of persistent HEV infection in human hepatocytes in which HEV replicates efficiently. This HEV cell culture system is based on differentiated HepaRG cells infected with an isolate of HEV-3 derived from a patient suffering from acute hepatitis E. Efficient replication was maintained for several weeks to several months as well as after seven successive passages on HepaRG naïve cells. Moreover, after six passages onto HepaRG, we found that the virus was still infectious after oral inoculation into pigs. We also showed that ribavirin had an inhibitory effect on HEV replication in HepaRG. In conclusion, this system represents a relevant and efficient in vitro model of HEV replication that could be useful to study HEV biology and identify effective antiviral drugs against chronic HEV infection.

## 1. Introduction

Hepatitis E virus (HEV) is a single stranded positive RNA virus and a member of the *Hepeviridae* family within the *Orthohepevirus* genus [1]. Its 7.2 kb genome codes for three open reading frames (ORFs) [2]. *ORF1* codes for a non-structural polyprotein containing several putative functional domains: a methyltransferase (Met), a papain-like cysteine protease (PCP), a proline-rich hypervariable region (HVR), a macrodomain (or X domain), a helicase (Hel) and a RNA-dependent RNA polymerase (RdRp) [3]. *ORF2* codes for the capsid protein and a secreted form of glycosylated ORF2 [4]. *ORF3* encodes a small phosphoprotein involved in viral release and other functions. In addition, *ORF4*, an overlapping reading frame within *ORF1*, has been identified recently in HEV genotype 1 only and codes for a viral protein involved in viral replication [5]. Four main genotypes of HEV are able to infect humans. Genotypes 1 and 2 (HEV-1 and HEV-2) infect only humans and are present in Asia, the Middle East, Africa and Central America. These genotypes are transmitted primarily via the faecal–oral route, through the consumption of contaminated water in areas with poor sanitation. In contrast, genotypes 3 and 4 (HEV-3 and HEV-4) are zoonotic pathogens. They are found worldwide in several animal species (mainly pig, wild boar and deer) and can be transmitted via the consumption of infected meat or direct contacts with infected animals [6,7]. In addition, a case of human infection with camelid HEV-7 has also been reported [8]. In most human cases, HEV infection is asymptomatic or causes an acute hepatitis that is self-limited. However, fulminant hepatic failure can occur in patients with underlying chronic liver disease, in the elderly and, for HEV-1, in pregnant women. Chronic cases of HEV-3 and HEV-4 infection have also been reported in immunocompromised patients who have received solid-organ transplants, with human immunodeficiency virus infection or with hematologic cancers under chemotherapy [9,10,11]. Chronic infection has also been reported in a liver transplant recipient infected with HEV-7 [8]. Such infections can evolve rapidly to cirrhosis, end-stage liver disease, the need for liver transplantation and loss of liver graft in transplant recipients [12]. More recently, extrahepatic manifestations including renal, pancreatic and neurological disorders have been linked to HEV infection [13]. With the exception of China, no vaccine is available. No specific treatment against HEV infection has been approved yet, even if several reports have described the successful use of ribavirin to treat cases of chronic hepatitis E. Unfortunately, side effects can occur with this drug (anaemia) and it cannot be used in pregnant women [14]. Moreover, clearance of the virus can sometimes fail or viral recurrence can occur after treatment is stopped [14,15]. A study also suggests that ribavirin treatment can causes mutations in the HEV genome [16]. However, the emergence of such mutations in the viral polymerase has not been associated so far with drug resistance or failure to clear the virus [16,17,18]. In a few cases, pegylated interferon-α has also been used to treat chronic hepatitis E. However, this drug causes significant adverse effects and organ rejection in transplant patients that represent the majority of patients with chronic hepatitis E [19]. Therefore, there is a real need to better understand chronic HEV infection and develop models that will contribute to the development of an effective treatment. Several in vitro models for HEV including hepatoma cells (PLC/PRF/5, Huh-7, HepG2/C3A), primary human hepatocytes (PHH), hepatocyte-like cells (HLCs), stem-cell derived hepatocytes and non-liver cells (intestinal Caco-2, alveolar A549, placental-derived cells JEG-3 and primary small intestine epithelial cells) have been developed [20,21,22,23]. However, for decades, these cell culture systems did not allow replication of HEV at high levels. It is only recently that optimised cell models have been developed and high HEV viral loads detected in HepG2/C3A, A549, PLC/PRF/5 and Huh-7-Lunet BLR cells [24,25,26]. Increased viral replication has also been reported in hepatoma cells expressing HEV-1 ORF4 [27]. Our group previously developed an in vitro system of HEV infection in human HepaRG cells embedded in 3D Matrigel [28]. HepaRG cells are able to differentiate into both cholangiocyte and hepatocyte-like cells and are considered as a good alternative system to primary human hepatocyte (PHH) culture to investigate host–pathogen interactions [29]. Earlier, using this system, up to 1 × 10^3^ HEV genome equivalents (GE) per ml of supernatant were detected in the supernatant of infected cells (multiplicity of infection (MOI) of 0.5) 32 days after infection with an HEV-3 isolate originating from swine faeces [28]. In the present study, we aimed to develop an optimised HepaRG model in which HEV replicates more efficiently. We established a HEV cell culture system based on differentiated HepaRG cells infected with an isolate of HEV-3f derived from a patient suffering from acute hepatitis E. Using a faecal suspension with a high viral load, efficient replication was obtained (with concentrations reaching up to 1 × 10^9^ GE/mL of supernatant) that could be maintained for several weeks to several months. Moreover, the virus was passaged several times in HepaRG without loss of replication efficiency and remained infectious in pigs after six passages. We also showed that ribavirin had an inhibitory effect on HEV replication. In conclusion, this cell culture system of persistent HEV replication in differentiated HepaRG represents a relevant in vitro system to study HEV infection in human hepatic cells. 

## 2. Materials and Methods 

### 2.1. Cell Culture

Undifferentiated human HepaRG^TM^ cells were purchased from BIOPREDIC International (Saint-Grégoire, France). Cells were grown in William’s E medium with GlutaMAX^TM^ (Thermo Fisher Scientific, Waltham, MA, USA) supplemented with 10% heat-inactivated foetal calf serum (FCS), 5 μg/mL insulin (Sigma-Aldrich, Saint-Louis, MO, USA), 5 × 10^−5^ M hydrocortisone hemi-succinate (Sigma-Aldrich, Saint-Louis, MO, USA) and 100 IU/mL penicillin and 100 µg/mL streptomycin (growth medium). Cells were maintained at 37°C in 95% air/5% CO_2_. Confluent HepaRG monolayers were passaged every 2 weeks and medium was renewed every 2-3 days. For differentiation, HepaRG were seeded into 6-well plate (2 × 10^5^ cells/well) and cultured in growth medium for 2 weeks. Medium was then replaced by William’s E medium with GlutaMAX^TM^ (Thermo Fisher Scientific, Waltham, MA, USA) supplemented with HepaRG™ Differentiation Medium with antibiotics (ADD720C, BIOPREDIC International, Saint-Grégoire, France) (differentiation medium containing dimethyl sulfoxide (DMSO)) for 2 extra weeks and then maintained in growth medium for 1 to 2 days before infection. 

### 2.2. Virus 

Human faeces was collected from a French patient suffering from acute autochthonous hepatitis E of subtype 3f [30]. The faecal sample was resuspended in phosphate-buffered saline (PBS), centrifuged at 12,080 g and passed through a 0.22 μm filter to obtain a faecal suspension with a high viral load (1.4 × 10^9^ GE/mL). Viral load is expressed in genome equivalent, determined by reverse transcription-quantitative polymerase chain reaction (RT-qPCR) as RNA copies. The consensus sequence of the full-length genome of this human HEV isolate FR-HuHEVF3f is accessible in GenBank under accession number JN906974. Analysis of high-throughput sequencing data generated for the human faeces sample used as starting inoculum in this study has been published previously [30]. Forty-two single-nucleotide polymorphisms (SNPs) were identified, corresponding to 0.5% of the genome. The proportion of the HEV population displaying one particular SNP reached 20% [30].

### 2.3. Virus Inoculation and Passages

HepaRG cells were differentiated in six-well plates and infected for 24 hours with an HEV inoculum (faecal sample or supernatant) diluted in growth medium to a final volume of 1 mL. The viral suspension was then removed and cells were washed three times in PBS before adding 2 mL of growth medium. Every 2 to 3 days, one-half (1 mL) of the culture medium was replaced with fresh growth medium and infection maintained for up to 382 days (passage 2). Cells were not passaged during the duration of the infection. Six consecutive passages of the virus were then carried out onto fresh HepaRG cultures using this protocol. Details of the inoculum, MOI and length of infection used at each passage are presented in Table 1. Supernatants from infected cells used as inoculum were centrifuged prior to infection to remove cell debris. For ribavirin treatment, cells were infected for 12 to 13 days before addition of ribavirin (R9644, Sigma-Aldrich, Saint-Louis, MO, USA) into the supernatant to final concentrations of 0, 10, 50, 100 or 200 μM. Every 2 to 3 days, one-half (1 mL) of the culture medium was replaced with fresh growth medium containing the corresponding ribavirin concentrations and the cells treated for 21 to 30 days. Cells were then maintained for a further 13 to 14 days with one-half (1 mL) of the culture medium replaced with fresh growth medium without ribavirin every 2 to 3 days. 

### 2.4. Experimental Infections 

Five-week-old specific pathogen free (SPF) Large White piglets, free from HEV and anti-HEV maternal antibodies were used. Four pigs were kept as control and two pigs were orally inoculated with 10^8^ HEV RNA copies of a sixth passage of the human HEV-3f on HepaRG cells in a volume of 10 mL. Faecal samples were collected three days before inoculation and three times a week until the end of the experiment at 49 days post inoculation. Blood samples were collected before inoculation and once a week until the end of the experiment. The experiment was performed at the Agence Nationale de Sécurité Sanitaire, de l’environnement et du travail (ANSES)’s air-filtered level-3 biosecurity facilities in accordance with EU and French regulations on animal welfare in experiments, alongside a previously reported experimental infection [31]. This protocol was approved (referral 17-022) by the ANSES/Ecole nationale vétérinaire d’Alfort (ENVA)/Université Paris-Est Créteil (UPEC) ethical committee registered under number #16 (24 February 2017). 

### 2.5. RNA Extraction and Quantification of HEV RNA by TaqMan RT-PCR

Totals RNAs from pelleted cells were extracted using the RNeasy Minikit (Qiagen, Hilden, Germany) according to the manufacturer’s protocol. Viral RNAs were extracted manually from culture supernatants or faecal samples using the QIAamp Viral RNA Mini kit (Qiagen, Hilden, Germany) as previously described [32]. HEV RNA quantification was adapted from a previously described method [33]. The QuantiTec Probe RT-PCR kit (Qiagen, Hilden, Germany) was used according to the manufacturer’s instructions using 2 μL of RNA (template), 0.25 mM reverse primer (5′-AGGGGTTGGTTGGATGAA-3′), 0.1 mM forward primer (5′-GGTGGTTTCTGGGGTGAC-3′) and 5mM probe (FAM-TGATTCTCAGCCCTTCGC-MGB) as previously described [34]. A LightCycler 480 apparatus (Roche, Basel, Switzerland) was used for sample analysis. Reverse transcription was carried out at 50 °C for 20 min, followed by denaturation at 95 °C for 15 min. DNA was amplified with 45 cycles at 95 °C for 10 s and 58 °C for 45 s. Standard HEV RNA was obtained after in vitro transcription of a plasmid pCDNA 3.1 ORF2-3 HEV, as described previously [32] and used to generate standard quantification curves.

### 2.6. Immunoblot Analysis

Cells were lysed in lysis buffer (25 mM Tris HCl pH 8.8, 50 mM NaCl, 0.5% Nonidet P-40 and 0.1% sodium dodecyl sulphate supplemented with cocktails of protease and phosphatase inhibitors). Debris were removed by centrifugation at 16,000 g for 20 min at 4°C. Total protein concentration was then determined using a Micro BCA^TM^ Protein assay (Thermo Fisher Scientific, Waltham, MA, USA). Equal amount of protein lysate was heat-treated in loading sample buffer containing β-mercaptoethanol and analysed by 12% sodium dodecyl sulphate-polyacrylamide gel electrophoresis (SDS-PAGE). Proteins were then transferred to nitrocellulose membrane (Hybond-ECL, Amersham, GE healthcare LifeScience, Pittsburgh, PA, USA). Membranes were covered with blocking buffer (PBS containing 5% dry milk and 0.05% Tween-20) for one hour and then incubated with a mouse anti-HEV ORF2 antibody (MAB8002, Merck Millipore, Darmstadt, Germany) or a mouse anti-actin antibody (clone AC-40, Sigma-Aldrich, Saint-Louis, MO, USA) diluted in blocking buffer (1/500 and 1/2000 dilution, respectively). After several washes in PBS containing 0.05% Tween-20, a horseradish peroxidase-conjugated anti-mouse secondary antibody (Thermo Fisher Scientific, Waltham, MA, USA) was added (dilution 1/5000 in blocking buffer). An enhanced luminol-based chemiluminescent detection system was used to detect bound antibodies.

### 2.7. Immunostaining and Fluorescent Microscopy

Cells were seeded on 24-well plates (Ibidi μ-plates, BioValley, Marne la Vallee, France) and fixed with 4% paraformaldehyde in PBS. Cells were permeabilized with 0.2% Triton X-100 in PBS and incubated in blocking buffer (0.5% BSA in PBS). The monoclonal antibody directed against HEV ORF2 (MAB8002, Merck Millipore, Darmstadt, Germany) diluted (1/100) in blocking buffer was then added for 1 h at room temperature. Cells were then washed several times in PBS and DyLight^TM^ 550 anti-mouse secondary antibodies (Thermo Fisher Scientific, Waltham, MA, USA) were used to detect bound primary antibodies. Nuclei were stained with 4,6- diamidine-2-phenylindole dihydrochloride (DAPI) (Sigma-Aldrich, Saint-Louis, MO, USA) diluted in PBS. Microscopy was carried out with an Axio observer Z1 fluorescent microscope (Zeiss, Oberkochen, Germany) and images were acquired using the Zen 2012 software.

### 2.8. HEV Enzyme-Linked Immunosorbent Assay (ELISA)

HEV antibodies were detected using the HEV ELISA 4.0 V kit (MP Diagnostics, Illkirch, France) as described previously [35]. This sandwich ELISA allows the detection of IgG, IgM and IgA using a proprietary recombinant antigen that is highly conserved between HEV strains and is based on the ORF2.1 fragment (corresponding to amino acids 394 to 660 of the capsid protein) [36]. Samples were considered positive when the optical density at 450 nm was higher than the mean obtained for negative controls + 0.3.

### 2.9. Cell Viability Test

HepaRG cells were seeded into a 96-well plate (6.4 × 10^3^ cells/well) and differentiated as described above. Ribavirin (R9644, Sigma-Aldrich, Saint-Louis, MO, USA) was then added to the supernatant to a final concentration of 0 to 400 μM. Every two to three days, one-half (1 mL) of the culture medium was replaced with fresh growth medium containing the corresponding ribavirin concentrations and the cells treated for 12 days. Cells were then lysed and cell viability was determined using the CellTiter-Glo^®^ luminescent cell viability assay (Promega, Madison, WI, USA) according to the manufacturer’s recommendations. This assay is based on ATP quantification as indicator of metabolically active cells.

### 2.10. Whole Genome Sequencing and Sequence Analysis

RNA from 200 μL of filtered supernatant of HepaRG infected at passage 7, day 121 post infection, was extracted using the QIAamp Viral RNA Mini kit (Qiagen, Hilden, Germany) as previously described [28] with the exception that 20 μg/mL linear acrylamide RNA (Thermo Fisher Scientific, Waltham, MA, USA) was used instead of carrier RNA and RNA was eluted in 30 µL RNAse-free water. The cDNA libraries were prepared using Ion Total RNA-Seq kit v2 and sequenced on an Ion Proton instrument with an Ion chip Kit V3 (Life Technologies, Carlsbad, CA, USA). The reads were cleaned with the Trimmomatic [37] 0.36 software. The mean read length was 79 nucleotides. Then a Bowtie2 [38] (version 2.2.5) alignment was performed with down-sampled reads on local nucleotide database to estimate HEV read numbers. The references with the highest number of matching reads was used for an alignment with Burrows-Wheeler aligner (bwa) [39] (0.7.15-r1140). The reads were then down-sampled to fit a global coverage estimation of 80 x and were submitted to the SPAdes (3.10.0) for a de novo assembly [40]. The de novo contigs were then submitted to MEGABLAST [41] on a local nt database, and the best match was then selected for a bwa (0.7.15-r1140) alignment [39]. The de novo assemblies and the alignment on the reference were then compared and the strict identities of the de novo and aligned sequences were assessed for validation of the final sequence. In case of discrepancies of the de novo and the aligned sequences, assessment of the bam (visualized with IGV) was performed for a manual curation of the sequence. A SNP calling was performed with Varscan [42]. Sequences obtained for this study are available in GenBank: Bioproject # PRJNA685626. The consensus sequence of the full-length genome of HEV after 7 passages in cell culture (HEV3f-P7H) is accessible in GenBank under accession number MW383253. 

### 2.11. Phylogenetic Analysis

A blast analysis identified that the sequences most closely related to FR-HuHEVF3f were HEPAC-15, TLS09-3, SW8a24 Spain and JAO-SpaTok12. Alignments were performed with ClustalW using the complete or near-complete genome sequences of these HEV-3f sequences as well as the reference sequences of the different genotypes of HEV (HEV-1 to -8) and subtypes of HEV-3 [43]. A phylogenetic tree was then constructed by using the Maximum Likelihood method and Tamura-Nei model with a bootstrap of 1000 replicates. The initial tree was obtained using the Neighbor-Joining method. This analysis was conducted using the MEGA X software (http://www.megasoftware.net, accessed on 5 February 2021).

### 2.12. Statistical Analyses

Unpaired *t* test was used to analyse the data. Differences were considered to be significant if the *p* value was < 0.05. 

## 3. Results

### 3.1. Culture of a Human HEV-3f Strain in Differentiated HepaRG Cells

In an attempt to develop a relevant and efficient system of HEV infection in vitro, we tested the ability of a HEV patient isolate to replicate in the human hepatic cell line HepaRG. High viral load inoculum (1.4 × 10^9^ GE/mL) was obtained from the stool of a patient infected with subtype 3f of HEV and suffering from acute hepatitis E (strain FR-HuHEVF3f, GenBank accession number JN906974) [30]. HepaRG cells were differentiated into hepatocytes and cholangiocytes via the addition of hydrocortisone and DMSO using a well-established protocol [44] and inoculated with the HEV faecal suspension at a MOI of 1 and 10 HEV GE per cell. The presence of viral RNA was then monitored by RT-qPCR for up to 95 days (Figure 1a). High levels of viral RNA were detected one day post-infection, reflecting viral overload from the inoculum. HEV RNA then decreased for the first 10 days to increase consistently after 2 weeks of infection and reached up to 2 × 10^8^ GE/mL of supernatant after 95 days of infection. 7.32 and 9.89 × 10^4^ GE/μg of total RNA were also detected intracellularly after 29 days of infection for MOI 1 and 10, respectively. No cytopathic effects (CPE) were observed. To confirm that the HEV RNA detected in the supernatant corresponded to infectious viral particles, the supernatant of infected HepaRG cells was then inoculated onto fresh differentiated HepaRG cells and this step repeated for six successive passages of the virus (Table 1 and Figure 1b,c). Increased viral RNA was detected in the supernatant of cells infected at these different passages, reaching up to 1.18 × 10^9^ GE/mL. These results clearly indicate that the human HEV-3f strain replicates in HepaRG cells and that infectious particles are released in the supernatant. No CPE were observed at the different passages over the entire period of infection. Remarkably, cells could be maintained for several months to more than a year without affecting the viral load detected in the supernatant of infected cells (Table 1 and Figure 1b). Our cellular model then allowed us to produce viral stocks with high concentrations. Interestingly, after 7 passages, higher concentrations of virus were released in the supernatant of infected cells during the first months of infection and the time interval between inoculation and maximum virus yield was reduced (Table 1 and Figure 1c). 

### 3.2. Intracellular Detection of HEV in Differentiated HepaRG Cells

The kinetics of intracellular levels of HEV RNA was also measured in differentiated HepaRG (Figure 2a). Increased viral RNA was detected in infected HepaRG cells, ranging from 1.2 × 10^4^ at 1 day post-infection to 2.6 × 10^7^ HEV GE/μg total RNA at 72 days post-infection. Moreover, expression of the HEV capsid, ORF2, was also detected in infected human hepatocyte cells by immunoblotting (Figure 2b) and immunofluorescence (Figure 2c) but not in mock-infected cells. 

### 3.3. Ribavirin Inhibits HEV Replication in Differentiated HepaRG Cells

To further validate the replication of HEV-3f in differentiated HepaRG and the pertinence of the HepaRG system in studying HEV infection, the effect of ribavirin, a viral replication inhibitor, was studied. Previous studies have reported that this drug inhibits HEV replication in vitro and in patients with chronic hepatitis E infection [14,45,46]. First, we assessed ribavirin toxicity on differentiated HepaRG cells and found that concentrations ≤ 200 μM had no effect on HepaRG cell viability (Figure 3a). We then assessed the effect of different concentrations of ribavirin on HEV replication (Figure 3b–d) and found that at concentrations higher ≥ 50 μM, HEV genome in the supernatant decreased significantly and was no longer present 21 to 30 days after treatment. Moreover, treatment with 10 μM of ribavirin reduced HEV RNA concentration in the supernatant by more than 80% after 23 days of treatment (Figure 3b). HEV genome present in the intracellular compartment was also reduced significantly for all concentrations used. These results then show that ribavirin inhibits HEV replication in this cell culture system and that this model of persistent HEV infection could be useful to test antiviral drugs for the treatment of chronic hepatitis E.

### 3.4. HEV Remains Infectious In Vivo in Pigs after Several Passages in HumanCcells

To determine whether the virus released in the supernatant of infected cells after several passages on HepaRG is still infectious in vivo, two pigs were inoculated orally with 1 × 10^8^ HEV GE per animal in a volume of 10 mL. HEV excretion in the faeces was then monitored for up to 49 days (Figure 4a). Virus was detected in one out of the two pigs at 22 and 23 days post-infection at a concentration of 1.54 × 10^4^ and 3.28 × 10^5^ GE/g, respectively. Seroconversion was also monitored and anti-HEV antibodies were detected in both pigs (Figure 4b). We can, then, not exclude that both pigs were infected. It is possible that HEV RNA excretion was not detected in the faeces of one of the two infected pigs because inhibitors were present in the faeces or because excretion was too short or below the detection limit. Four control pigs were also inoculated in parallel with PBS. No HEV excretion nor anti-HEV antibodies were detected in these animals. These data suggest that virus particles excreted by HEV-infected HepaRG cells are still infectious in vivo even after six successive passages of the virus. 

### 3.5. FR-HuHEVF3f Has an Insertion within the ORF1 HVR

Phylogenetic analyses based on the analysis of complete genome sequences of several HEV genotypes and HEV-3 subtypes showed that FR-HuHEVF3f clusters closely together with strains isolated in human samples in France (HEPAC-15-2010 and TLSO9) [47,48] and Japan (JAO-SpaTok12) and in swine in Spain (SW8a24_Spain) (Figure 5a). These five strains contain an insertion of 29 amino acids within the HVR region of ORF1 (Figure 5b). For FR-HuHEVF3f, the amino acid sequence of this insertion is 65% (19/29 amino acids) identical to the adjacent sequence (Figure 5c), suggesting that a duplication occurred within the HVR. 

### 3.6. Evolution of HEV Genome Sequence after Seven Passages in Human Hepatocytes 

Next, we investigated genetic variability of HEV at passage 7, day 121 post infection (Figure 1b), using whole genome sequencing. In the sample analysed, 25 mutations had been selected along two of the three ORFs of the HEV genome (Table 2). Of these 25 mutations, eight mutations are non-synonymous, including seven in the open reading frame 1 (ORF1) and one in the viral capsid (ORF2). 

## 4. Discussion

For decades, the difficulty of growing HEV in cell culture has considerably delayed our understanding of HEV biology and the development of effective control measures against the virus. As described in the introduction, many efforts have been made in the past 12 years to develop efficient in vitro models of HEV replication. However, the cell lines used are not the main target of HEV (A549), secrete hepatitis virus B surface antigen (PLC/PRF/5) [49] or have a deficient interferon signaling pathway (HepG2 and Huh-7) [50]. Primary human hepatocytes [51,52] and stem cell-derived hepatocytes [53,54,55] support HEV replication and represent more relevant systems to study the virus, but they are more difficult to grow, less available, display more variability and/or require complex differentiation and handling protocols [20,21]. Several cDNA clones and replicon systems have also been developed that are able to replicate efficiently but that are less relevant physiologically than clinical isolates [21]. Hence, there is a real need to develop accessible, efficient and physiologically relevant hepatocyte systems for HEV infection [20,21]. 

In the present study, we developed a system of persistent HEV infection in human hepatocyte-like HepaRG cells that express the main markers of in vivo hepatic progenitors [44,56]. Differentiated HepaRG cells that display both hepatocyte-like and cholangiocyte-like epithelial phenotypes were infected with a human fecal suspension containing a high viral load of HEV-3f. The virus was then successfully passaged into fresh cells and HEV RNA concentration of up to 1 × 10^9^ copies/mL of culture supernatant were obtained and maintained for several months. HEV ORF2 was also detected by immunofluorescence and immunoblotting within infected cells after serial passages of the virus. These results clearly show that HEV replicates efficiently in differentiated HepaRG cells for prolonged time and that infectious progeny viruses are released in the supernatant. This was confirmed by experimental infection showing that the virus is still infectious in pigs after six passages in these cells.

HepaRG were previously used in a few studies aimed to investigate HEV. However, low replication levels were obtained when Matrigel-embedded HepaRG were infected with a swine HEV-3f isolate (up to 1 × 10^3^ HEV GE/mL supernatant after 32 days of infection) [28] or a cDNA clone system (full-length Kernow-C1 p6 HEV) was used to electroporate or infect non-differentiated HepaRG cells [57,58]. The system presented in this study is different as cells were differentiated via the addition of DMSO and a human HEV-3f isolate was used. This cell/virus combination allowed us to detect efficient replication for prolonged time and to successfully passage the virus. HepaRG cells differentiated upon exposure to DMSO are already used as a relevant model to study and develop antiviral therapies against HBV [44,59] and HCV [60] infection. They express a similar pattern of functional TLR/RLR as primary human hepatocytes and subsequently are a good surrogate model to study interactions between pathogens and the hepatocyte innate immune system [29]. They are also a good model for drug discovery and have been widely used for drug metabolism and toxicity assessments [61,62,63]. Indeed, HepaRG cells show similar morphology, metabolic activity and expression profiles as human hepatocytes and can maintain liver cell functions and drug-metabolizing enzymes better than other hepatoma cells [64]. Here, we show that ribavirin is able to inhibit HEV replication in differentiated HepaRG, as previously reported in vitro and in vivo [14,45,46]. Treatment with 10 μM ribavirin greatly reduced HEV RNA level in the supernatant of infected cells and HEV RNA clearance was achieved with concentration equal or superior to 50 μM. Similar concentrations of the drug were shown to be effective in other in vitro models. In Huh7.5 transfected with different HEV replicons, inhibitory concentrations 90 (IC90) of 50 to 64 μM were determined after 42 h treatment [65]. In infected PLC/PRF/5 cells, it was shown that HEV growth was suppressed by 160 μM ribavirin but not by concentrations equal or inferior to 80 μM even after 28 days of treatment [66]. In Huh7 cells infected with the Kernow-C1 p6 virus, treatment with 100 μM ribavirin for 20 days led to a 4-log10 reduction in viral titer without complete clearance of the virus [45]. In addition, HEV replication was inhibited by more than 95% after treatment with 100 μM ribavirin in human primary intestinal cells infected by different strains of HEV [22]. Thus, the system of persistent HEV infection presented in this study is a promising tool to identify antiviral drugs effective against HEV replication in the context of prolonged HEV infection.

In our cell culture system, cells were infected with a virus isolate recovered from a patient suffering from autochthonous acute hepatitis E. Analysis of its sequence revealed the presence of an insertion of 87 nucleotides (29 amino acids) within the HVR of the genome corresponding to the duplication of the adjacent HVR sequence. Similar 87 nucleotide duplication have already been found in other HEV-3 field isolates as well as duplication of 69 and 39 nucleotide fragment belonging to this 87 nucleotide region [47,48,67,68]. Duplications of 69 and 87 nucleotides were only found in the HEV-3f subtype whereas insertions of up to 18 or 39 nucleotides were found in HEV-3a and HEV-3e isolates [47,48,67,68]. Interestingly, in-frame insertions at similar location within the ORF1 HVR have been reported for several clinical strains of HEV able to replicate in cell culture (Table 3). Some of these insertions were shown to enhance viral replication and provide growth advantage to the virus in cell culture. Insertions within the HVR region have also been reported in patients suffering from chronic hepatitis E (insertion of 75 nucleotides of human inter-alpha-trypsin inhibitor heavy chain 2 (ITI-H2) sequence [47] or duplication of HVR sequence (258 nucleotides) and insertion of RdRp sequence (24 nucleotides) [18]). The function of the HVR, also known as the polyproline region (PPR), is still unclear. Some studies have suggested that it is an intrinsically disordered region that plays a structural role by acting as a hinge or spacer [3,67,68]. Analysis of HVR sequences from different genotypes suggest a role of this proline-rich region in HEV replication regulation and evolution [67,68]. Moreover, a study has shown that mutants lacking part of the HVR are still infectious but replicate less efficiently in vitro and are attenuated in vivo when a large part is deleted [69,70]. In this study, the 87 nucleotide insertion was already present within the HVR region of the HEV isolate used to infect cells at the first passage. This might have facilitated replication of the virus onto differentiated HepaRG. Interestingly, after seven passages of the virus in our cell culture system, higher concentrations of HEV RNA were detected in the supernatant of infected cells for several months after infection in comparison to the first and second passage. Moreover, the time interval between inoculation and maximum virus yield was reduced, although similar or lower MOI was used at passage 7 (Table 1 and Figure 1c). This suggests adaptation of the virus to grow in cell culture. Whole-genome sequencing of the virus after seven passages showed the presence of 25 mutations including eight non-synonymous mutations. Of these, seven were found in ORF1 (two in PCP, two in the region between PCP and HVR, one in HVR, one in X and one in RdRp), one in the viral capsid and none in ORF3. None of these selected mutations were detected, even in a minority, in the original fecal inoculum by high-throughput sequencing [30]. It is possible that, through persistent infection and passages, one or several of these mutations have improved the replication fitness of the virus in HepaRG cells while maintaining the capacity to remain infectious in vivo and cross host species barriers. In a previous experimental infection, no nucleotide mutation was detected over the full-length consensus sequence of the human FR-HuHEVF3f isolate after transmission to pig, suggesting that interspecies transmission is not modulated by host-specific mutations [30]. After seven passages in HepaRG, the higher number of point mutations was found in the viral sequence coding for PCP (amino acids 433-592) and HVR (amino acids 707-817) and the region situated between these two domains. It is possible that this part of the HEV genome is involved in interaction with the host cell. Interestingly, PCP has already been shown to interfere with the interferon system [71,72,73] and has deISGylation activity that may be critical for counteracting host antiviral pathways [74]. Moreover, this domain of HEV ORF1 may act as a protease implicated in ORF1 processing and replication of the virus genome [75]. Other studies aimed at identifying mutations derived from cell culture adaptation have also found that non-synonymous point mutations occur frequently in the HVR region (Table 3). Interestingly, the HVR region was also shown to be the region the most prone to non-synonymous mutation in a patient with chronic hepatitis E over a 4 year period (19 nonsynonymous mutations including 3 in Met, 5 in Y, 5 in HVR + 1 insertion, 2 in X, 1 in Hel, 2 in RdRp and 2 in ORF2) [47]. As discussed above, these findings also suggest that HVR plays a role in cell culture adaptation in the absence of major selective pressure. Nevertheless, further analyses are needed to study the adaptive evolution of the FR-HuHEVF3f isolate after passages in differentiated HepaRG cells and to determine whether one or a combination of the eight non-synonymous mutations identified in this study have an impact on viral fitness. This latter point requires the use of reverse genetics and the construction of full-length infectious cDNA clones.

In conclusion, our system of persistent HEV infection in differentiated HepaRG cell culture provide a useful tool to study the virus, identify host–virus interactions and putative mutations within the viral genome than can occur in vitro in the context of prolonged hepatitis E infection. This might then contribute to a better understanding of the underlying mechanisms of chronic hepatitis E infection and to the characterization of antiviral drugs effective against HEV infection. In addition, this infection system could be useful to generate viral stocks for in vitro and in vivo infection. Studies are also ongoing to evaluate the utility of our cell culture model to isolate other HEV strains and to detect the presence of infectious particles in clinical and field samples (food product, environment).

## Figures and Tables

**Figure 1 viruses-13-00406-f001:**
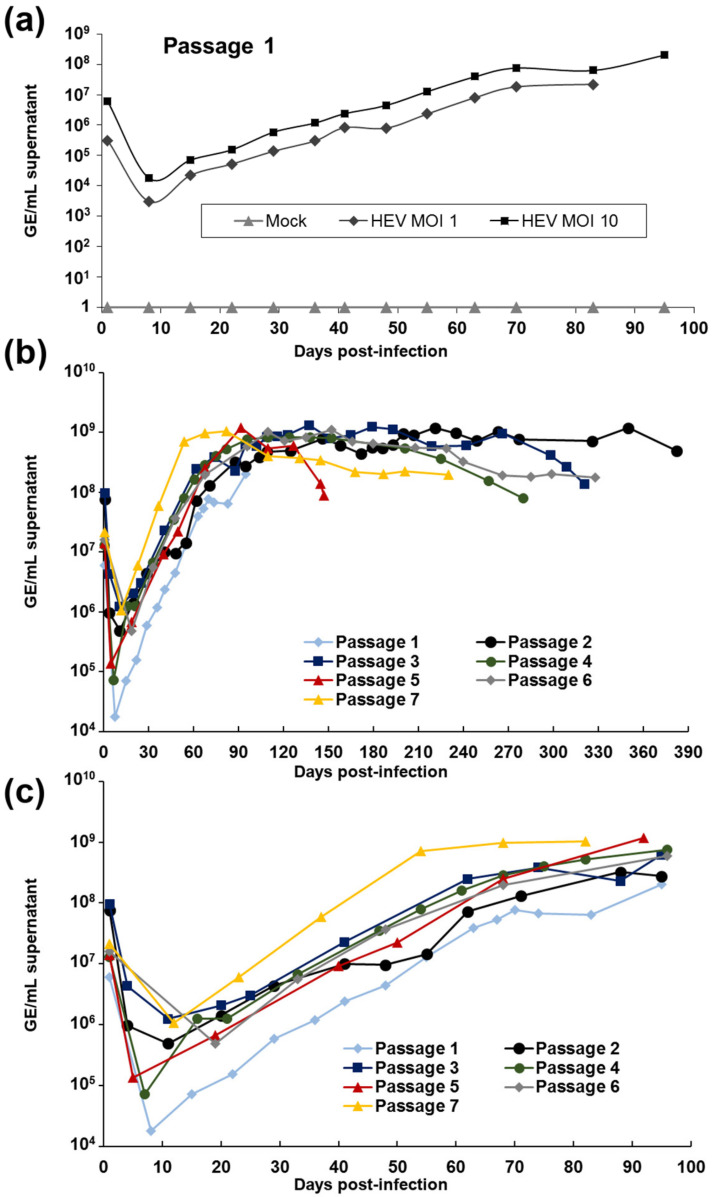
Replication of a human HEV-3f strain in differentiated HepaRG cells. (**a**) Inoculation of differentiated HepaRG with a faecal suspension of HEV at a multiplicity of infection (MOI) of 1 and 10. The HEV genome present in supernatants of the cell cultures was quantified using RT-qPCR at different days post-infection. (**b**,**c**) Serial passages of HEV strain in HepaRG. Viral concentrations in the supernatant are presented for the complete duration (**b**) and the 100 first days of infection (**c**).

**Figure 2 viruses-13-00406-f002:**
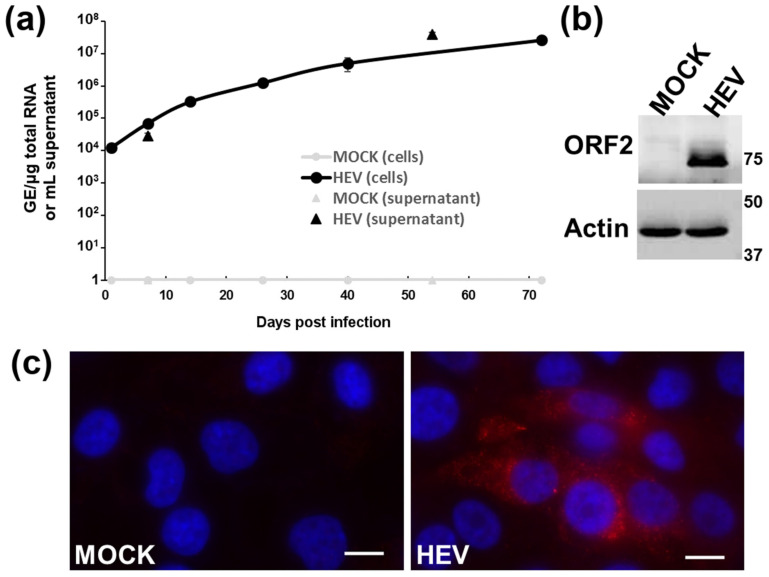
Detection of Hepatitis E virus (HEV) replication (**a**) and open reading frame (ORF)2 expression by immunoblotting (**b**) and immunofluorescence (**c**) within HEV-infected HepaRG cells. (**a**) Differentiated HepaRG cells were infected with the HEV-3f strain (passage 7) at a multiplicity of infection (MOI) of 10. The HEV genome present in cells and in the supernatants was quantified using RT-qPCR at different days post-infection. Results presented are the mean of triplicate samples (± standard deviations) (3 parallel infections). (**b**) Differentiated HepaRG were infected with HEV-3f (passage 6) at a MOI of 5 and cell lysates harvested 128 days post-infection. (**c**) Differentiated HepaRG cells were infected with HEV-3f (passage 7) at a MOI 50 for 37 days. Scale bars: 10 μm.

**Figure 3 viruses-13-00406-f003:**
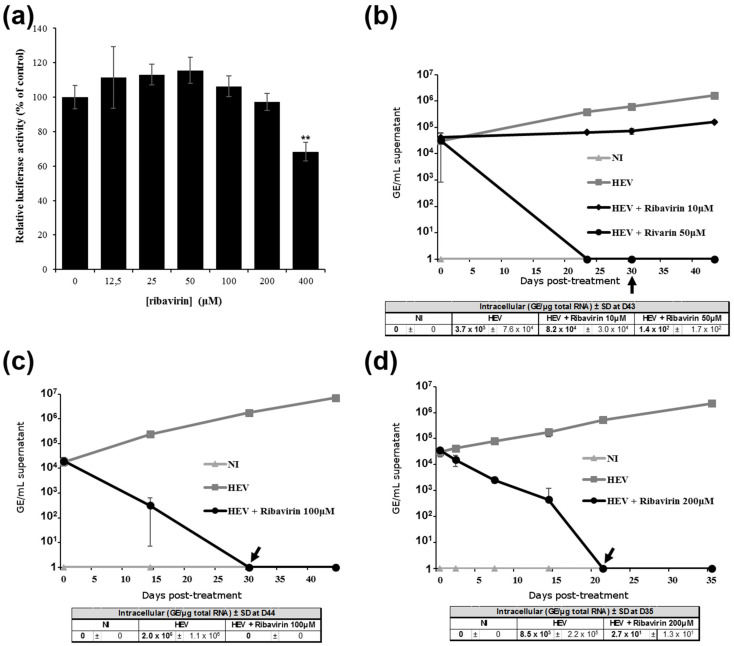
Inhibition of HEV replication by ribavirin. (**a**) Ribavirin cytotoxicity on differentiated HepaRG cells. Differentiated HepaRG were incubated for 12 days with up to 400 μM of ribavirin. Cells were then lysed and cell viability determined using a luminescent-based assay. Luciferase activities are expressed as percentage relative to non-treated cells. Mean of triplicate samples (± standard deviations). **, *p* < 0.005 compared to untreated cells (unpaired *t*-tests). (**b–d**) Effect of ribavirin on HEV genome release in the supernatants of differentiated HepaRG cells infected with HEV. Cells were infected for 12 to 13 days with HEV (faecal suspension) at a MOI of 2 before addition of 10 (**b**), 50 (**b**), 100 (**c**) or 200 (**d**) μM of ribavirin. Addition of drugs was stopped 21 (**d**) to 30 (**b**,**c**) days post-treatment as indicated by a black arrow. The HEV genome present in the supernatants of HepaRG cells was quantified using RT-qPCR at different days post-treatment. Tables show HEV concentrations in the intracellular compartments at the end of the experiment. Mean of triplicate samples (± standard deviations (SD)) are shown. NI = non-infected; D = days.

**Figure 4 viruses-13-00406-f004:**
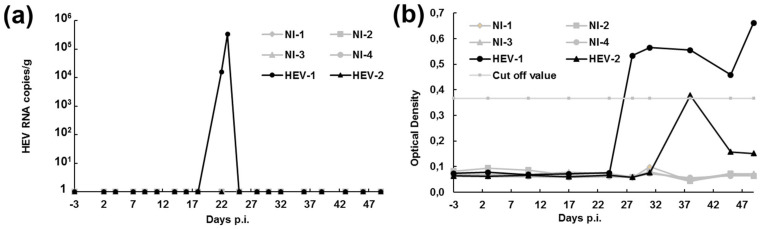
Experimental infection of two pigs with HEV harvested after six passages on HepaRG. (**a**) Kinetics of HEV excretion in the faeces quantified by RT-qPCR and expressed in number of HEV RNA copies/g of faeces. (**b**) Kinetics of HEV seroconversion. The presence of anti-HEV antibodies in serum was analysed by ELISA. NI = non-infected; p.i. = post-infection.

**Figure 5 viruses-13-00406-f005:**
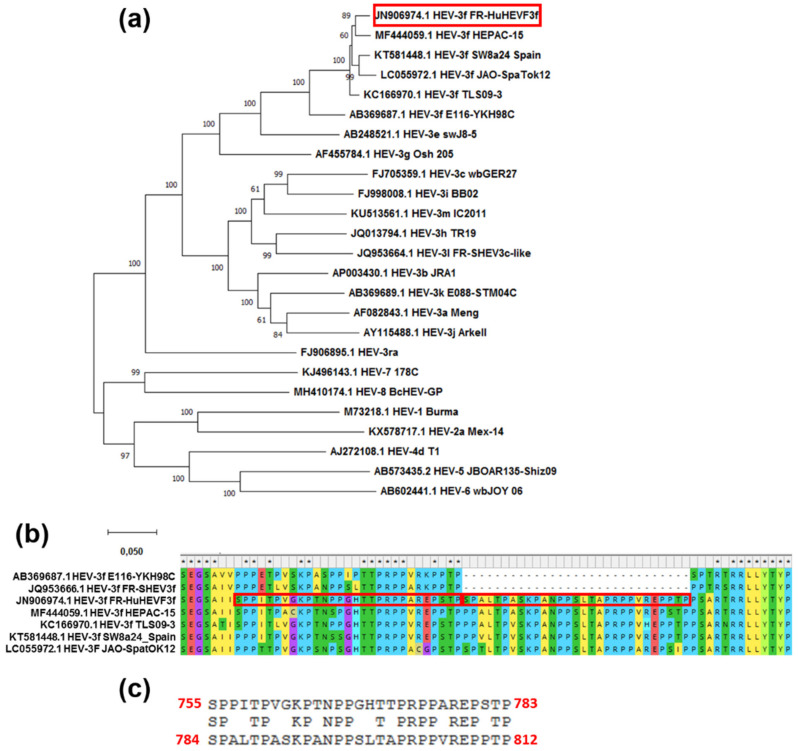
Duplication of a 29 amino acid sequence within the HVR of FR-HuHEVF3f ORF1. (**a**) Phylogenetic tree of 25 HEV complete or nearly complete sequences. This analysis includes FR-HuHEVF3f, different referenced genotypes of HEV (HEV-1 to -8) and subtypes of HEV-3 as well the 4 HEV-3f strains shown to be the most closely related to FR-HuHEVF3f in a blast analysis (HEPAC-15, TLS09-3, SW8a24 Spain and JAO-SpaTok12). The tree was constructed by using the Maximum Likelihood method and Tamura-Nei model based on the Clustal W alignment in MEGA X software. Bootstrap values were obtained from 1000 replicates. The GenBank database accession number of these complete sequences is indicated. FR-HuHEVF3f is indicated by a red box. Scale bar represents nucleotide substitutions per site. (**b**) Alignment of FR-HuHEVF3f with other HEV-3f strains revealed the presence of a duplicated 29 amino acid region in the hypervariable region (HVR) domain of ORF1 as indicated by red boxes. (**c**) Alignment of the duplicated sequence. The amino acid position within FR-HuHEVF3f ORF1 is indicated in red.

**Table 1 viruses-13-00406-t001:** Consecutive passages of HEV in HepaRG cells. The origin of the inoculum, the estimated multiplicity of infection (MOI) used as well as the length of infection are indicated for each passage. D = days.

Passage #	Inoculum	MOI (GE/Cell)	Inoculum Added (GE/Well)	Infection Length	Viral Load Around D+21 (GE/mL Supernatant)	Viral Load Around D+100 (GE/mL Supernatant)	Viral Load at the Last Time Point (GE/mL Supernatant)
1	Human faecal sample	10	2.1 × 10^7^	D+95	D+22: 1.55 × 10^5^	D+95: 2.02 × 10^8^	D+95: 2.02 × 10^8^
2	Supernatant 1st passage (D+91)	25	7.5 × 10^7^	D+382	D+20: 1.41 × 10^6^	D+95: 2.75 × 10^8^	D+382: 4.93 × 10^8^
3	Supernatant 2nd passage (D+84)	21	1 × 10^8^	D+321	D+20: 2.06 × 10^6^	D+95: 6.23 × 10^8^	D+321: 1.35 × 10^8^
4	Supernatant 3rd passage (D+41)	8	1.2 × 10^7^	D+280	D+21: 1.25 × 10^6^	D+96: 7.59 × 10^8^	D+280: 7.93 × 10^7^
5	Supernatant 4th passage (D+133)	10	3.2 × 10^7^	D+147	D+19: 6.72 × 10^5^	D+92: 1.18 × 10^9^	D+147: 8.73 × 10^7^
6	Supernatant 5th passage (D+126)	10	4 × 10^7^	D+328	D+33: 5.59 × 10^6^	D+96: 5.87 × 10^8^	D+328: 1.77 × 10^8^
7	Supernatant 6th passage (D+98)	10	3.8 × 10^7^	D+230	D+23: 5.95 × 10^6^	D+82: 1.04 × 10^9^	D+230: 1.97 × 10^8^

**Table 2 viruses-13-00406-t002:** Nucleotide mutations and amino acid substitutions of HEV virus HEV3f-P7H recovered at passage 7 in comparison to initial strain JN906974. Nucleotide positions are indicated with JN906974 base left and HEV3f-P7H base right. Amino acid (aa) substitutions and positions are indicated with JN906974 aa left and HEV3f-P7H aa right. ORF and putative motifs within ORF1 are indicated for each substitution. ORF1 unclassified region (Un) corresponds to the region between the PCP and HVR domains. Non-synonymous mutations are in bold. HEV3f-P7H Genebank accession # MW383253.

ORF	Nucleotide Positions	Amino Acid Substitutions	Motif
**ORF1**	G 538 T		
	T 628 C		
	T 925 C		
	T 1069 C		
	C 1114 T		
	**G 1499 C**	**G-493-R**	**Papain-like cysteine protease (PCP)**
	G 1534 A		
	**C 1794 T**	**A-591-V**	**PCP**
	**C 1821 T**	**A-600-V**	**Un**
	**C 2033 T**	**H-671-Y**	**Un**
	C 2197 T		
	**T 2246 C**	**W-742-R**	**HVR**
	G 2344 A		
	T 2494 C		
	**C 2630 T**	**H-870-Y**	**X**
	T 2674 C		
	**A 3974 G**	**T-1318-A**	**RNA-dependent RNA polymerase (RdRp)**
	C 4255 T		
	C 4621 T		
	T 4837 C		
	T 4918 C		
**ORF2**	C 6566 T		
	T 6740 C		
	**C 6756 T**	**L-500-F**	
	C 7205 T		

**Table 3 viruses-13-00406-t003:** HEV clinical isolates passaged several times in cell culture. Details of the name, origin of the isolate and infected cell line are given (pink fields). The presence of an insertion within the HVR is highlighted in green and its length, nature and position within the genome indicated. When determined, details of mutations identified within the HEV genome after several passages in cell culture are given (blue fields). nt: nucleotide; aa: amino acid.

Isolate	Origin (Patient)	Cell Line	Insertion				Point Mutations (Non-Synonymous)	Position of Non-Synonymous Mutation			Reference
			Length	Nature	Position	Cell Culture Adaptation		ORF1	ORF2	ORF3	
**LBPR-0379** (HEV-3a)	Liver transplant Chronic hepatitis E	HepG2/C3A	117 nt 39 aa	Human ribosomal protein S19	aa 741	Selected during passage	Not determined				[76]
**Kernow-C1** (HEV-3a)	HIV patient Chronic hepatitis E	HepG2/C3A	174 nt 58 aa	Human ribosomal protein S17	aa 750	Selected during passage Insertion shown to enhance growth	64 (16) + insertion of 58 aa in HEV ORF1 after 6 passages	G650A (un) V737A (HVR) N768S (HVR) I770V (HVR) L772S (HVR) P774S (HVR) W846S (X) N957S (X) Q1001H (Hel) F1459L (RdRp)	R2C V66I T483A M652S K653E	M69I	[77,78]
**47832** (HEV-3c)	Kidney transplant Chronic hepatitis E	A549	186 nt 62 aa	Duplication of adjacent HVR sequence (116 nt) and ORF1 3′ end (70 nt)	aa 750	-	25 (8) after 2 passages	F808S (HVR) F809S (HVR) A864V (HVR) D1201A (Hel) T1761I (RdRp)	V12M G73A	F108L	[79]
**JE03-1706** (HEV-3b)	Acute hepatitis E	PLC/PRF/5	None				18 (5) after 10 passages	A189V (Met) W741A (HVR) A1143V (Hel)		N73S P99L	[80]
**HEV0069** (HEV-3c)	Heart transplant Chronic hepatitis E	A549	None				19 (7) after 7 passages	FL462L (PCP) W741R (HVR)	P68S, E270V Y532D D625V	N73S P99L	[81]
**HE-JF5** (HEV-4c)	Fulminant hepatitis E	A549 PLC/PRF/5	None				10 (4) after 6 passages	F717L (HVR) V1617I (RdRp)	A565V K653E		[82]

## Data Availability

The references for data access are indicated in Section 2.

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
