# Peer review of "Characterization of a Cell Culture System of Persistent Hepatitis E Virus Infection in the Human HepaRG Hepatic Cell Line"

_viruses, 2021, doi:10.3390/v13030406_

Round 1

Reviewer 1 Report

Comments to the Authors

The authors present an HEV cell culture system based on the HepaRG cells. They successfully isolated a clinically relevant HEV-3f strain from faeces and passaged the strain seven times. High viral loads of 108-109 GE/ml were achieved after 6-10 weeks post inoculation depending on passage number. The strain could be cleared in vitro by adding 50-200 µM of ribavirin. Moreover, two pigs were inoculated with the sixth passage strain. HEV RNA was consequently detectable in one pig and only this pig developed a robust antibody response. The seventh passage strain was whole genome sequenced and compared to the strain derived from the patient. Several mutations were detected in the seventh passage strain, which were mainly located in ORF1. Meanwhile, several efficient HEV cell culture systems are established which efficiently produce high viral loads. The system presented by the authors may become one of these but I do have several comments:

Major comments:

  • Line: 77-79. The statement, “a relevant and effective cell culture system that would allow HEV to replicate at high titres is still lacking” is not true. Even the referenced paper in this context by Fu & Decker et al., 2019 mentions the latest HEV cell culture system in detail which produces high viral loads >109 GE/ml (PMID: 31141895). Therefore, the statement in line 313 “…titres never reached before” is also not correct. In addition, a more recent system using cDNA and cell culture derived particles (PMID: 31896581) also produces high viral loads in terms of FFU/ml. The authors are highly encouraged to include and properly cite the latest progresses on HEV cell culture system in their paper.
  • Did the authors also try to isolate other HEV strains to confirm the suitability of their cell culture system?
  • Infection of pigs was measured by solely testing faeces for HEV RNA. However, PCR can be inhibited when testing faeces. Did the authors include any nucleic acid isolation inhibition control in these experiments? Alternatively, pig serum should be screened for HEV RNA. Are there other explanations why the sixth passage-strain would only infect one out of two pigs?
  • The authors successfully passaged the HEV strain seven times and achieved remarkable viral loads of >108 GE/ml after every new passage. This makes the perfect basis for studying HEV mutation in vitro in more detail. Why did the authors only sequence the whole genome of seventh passage strain and not the ones of passage 1-6? This would add valuable information to the paper.
  • All three tables were not correctly inserted into the text and were therefore not fully reviewable. Moreover, explanation of abbreviations are missing at the end of most table and graph captions (e.g. Table 1: what does “J” stand for?; Figure 3: “NI” = not infected?; etc.).

Minor comments:

  • Line 13: HEV is only limited zoonotic, as HEV-1 and HEV-2 are restricted to humans (as later correctly stated by the authors in line 44-45).
  • Line 43-44: Not only four but also five HEV genotypes infect humans. The fifth is HEV-7 (PMID: 26551551)
  • Line 53: Only HEV-1 poses a severe threat to pregnant women.
  • Line 54: Chronification has also been reported for HEV-7.
  • Line 67-68: Ribavirin-induced HEV mutations do not confer resistance. The mutated virus is still sensitive to the drug (PMID: 27222534) and mutations do not affect viral clearance (PMID: 31793638).
  • Line 135-136: Were HEV positive cells passaged during the 382 days long maintenance or was the medium simply refreshed every 2-3 days?
  • Line 139: I guess Table 1 should be in section “2.3” rather than “2.6” (line 185).
  • Line 170-171: Since the MGB modification of the probe was used as stated in line 176, Garson et al., 2012 (PMID: 22871672) should be cited in this context.
  • Line 207: All ibidid µ-plates are made of polymer and only µ-slides are made of glass.
  • Line 226: Please add more details about the recombinant antigen (derived from which ORF? Length?)
  • Line 251: How long were the reads?
  • Line 289: Cholangiocytes, not choliangocytes.
  • Line 374: 1x108 GE per what?
  • Line 442-443: Why do the authors think, cDNA clones would replicate more efficiently than clinical isolates? Their data as well as data from Schemmerer et al., 2019 (PMID: 31141895) indicate that clinical isolates also replicate very efficiently.
  • Line 450-451: What was the exact viral load of the faecal suspension? In addition, HEV-3f instead of HEV-3 would be more informative to the reader.
  • Line 489-492: If I recall that correct, these insertions were only (or mainly?) common in subtype 3f strains. Could the authors provide more information about that in this context?
  • Line 556-558: Before stating the usefulness for detecting infectious particles in samples, the authors should evaluate the sensitivity of this cell culture system. Did the authors titrate the HEV-3f strain and were other HEV strains also isolated?
  • Figure 1: The authors state, they maintained the infection for up to 382 days but only show data for day 0-100. Please add the missing data.
  • Figure 5a: The tree should be recalculated using the latest HEV reference set by Smith et al., 2020 (PMID: 32469300).
  • Table 3: HEV strain 47832 is a subtype HEV-3c not HEV-3i. In addition, subtype information is missing for JE03-1706 and HE-JF5. Please add the correct subtype according to Smith et al., 2020 (PMID: 32469300).
  • Discussion: Are the ribavirin concentrations leading to viral clearance in this work similar to that of other publications?
  • Titres represent dilutions but in this work, viral loads are indicated as GE/ml and therefore as concentrations.

Author Response

Reviewer 1

The authors present an HEV cell culture system based on the HepaRG cells. They successfully isolated a clinically relevant HEV-3f strain from faeces and passaged the strain seven times. High viral loads of 108-109 GE/ml were achieved after 6-10 weeks post inoculation depending on passage number. The strain could be cleared in vitro by adding 50-200 µM of ribavirin. Moreover, two pigs were inoculated with the sixth passage strain. HEV RNA was consequently detectable in one pig and only this pig developed a robust antibody response. The seventh passage strain was whole genome sequenced and compared to the strain derived from the patient. Several mutations were detected in the seventh passage strain, which were mainly located in ORF1. Meanwhile, several efficient HEV cell culture systems are established which efficiently produce high viral loads. The system presented by the authors may become one of these but I do have several comments:

 We greatly thank Reviewer 1 for his/her comments and very useful suggestions. The manuscript has been modified to clarify all the points raised by reviewer 1. The modifications made to the manuscript are highlighted using the "track changes" feature in the word processing application. Responses to the reviewer comments are presented below in red color. The lines indicated refer to the revised manuscript including the “track changes”.

Major comments:

  • Line: 77-79. The statement, “a relevant and effective cell culture system that would allow HEV to replicate at high titres is still lacking” is not true. Even the referenced paper in this context by Fu & Decker et al., 2019 mentions the latest HEV cell culture system in detail which produces high viral loads >109 GE/ml (PMID: 31141895). Therefore, the statement in line 313 “…titres never reached before” is also not correct. In addition, a more recent system using cDNA and cell culture derived particles (PMID: 31896581) also produces high viral loads in terms of FFU/ml. The authors are highly encouraged to include and properly cite the latest progresses on HEV cell culture system in their paper.

We agree with Reviewer 1 that progresses have been made recently to provide more relevant and efficient cell culture system for HEV replication. The text has been changed to include details on the latest systems developed using cDNA, cell culture derives particles and field strains (line 91-102; reference 22-27).

In the statement in line 313, we meant, “titres never reached before” in differentiated HepaRG cells”. To avoid confusion, the text has been modified and “never reached before” deleted (line 365-366).

  • Did the authors also try to isolate other HEV strains to confirm the suitability of their cell culture system?

We have infected differentiated HepaRG cells with other HEV isolates and have detected replication with a HEV-1 virus isolated from the feces of a patient suffering from acute hepatitis E and with other HEV-3f isolates derived from the liver of an infected pig and a wild boar. The concentrations of HEV RNA detected in the supernatant of cells infected with these isolates were lower than the ones obtained with the HEV-3f strain described in this study and passages of the virus was not attempted or not successful. The goal of this manuscript is to characterize the system of HEV replication that involved differentiated HepaRG and the FR-HuHEVF3f isolate as this cell/virus combination allows efficient replication. The isolation of additional strains will be the subject of a future publication. A sentence has been added in the text to mention this point (line 657-663).

Infection of pigs was measured by solely testing faeces for HEV RNA. However, PCR can be inhibited when testing faeces. Did the authors include any nucleic acid isolation inhibition control in these experiments? Alternatively, pig serum should be screened for HEV RNA. Are there other explanations why the sixth passage-strain would only infect one out of two pigs?

We followed the same experimental design that has been validated in previous studies (Barnaud et al, 2012 (PMID: 22610436); Andraud et al, 2013 (PMID: 24165278); Salines et al, 2015 (PMID: 26048774 ); Salines et al, 2019a-b (PMID: 30599454 et PMID: 31213264)). We did not test pig serum for the presence of HEV RNA. We decided to test only feces as viral shedding is usually more prolonged than viraemia and that HEV RNA load is usually lower in serum than in feces (Salines et al, 2017). Nevertheless, it is possible that no HEV RNA excretion was detected in the feces of one of the two infected pigs because inhibitors were present in the faeces or because excretion was very short or low, below the limit of detection. We can then not exclude that both pigs were infected. This point is now discussed in the manuscript (line 438-442).

  • The authors successfully passaged the HEV strain seven times and achieved remarkable viral loads of >108 GE/ml after every new passage. This makes the perfect basis for studying HEV mutation in vitro in more detail. Why did the authors only sequence the whole genome of seventh passage strain and not the ones of passage 1-6? This would add valuable information to the paper.

We agree with reviewer 1 that this system provides a useful tool to study HEV mutations in vitro as mentioned in the discussion (line 651-657). The scope of this paper is to characterize an efficient system of HEV replication that was developed using the combination of differentiated HepaRG and the FR-HuHEVF3f isolate. Studying the molecular evolution and diversity of HEV after prolonged infection and passages would represent another study per se.  Studies on HEV mutation in vitro would require a reverse genetic system which is not available yet with the FR-HuHEVF3f strain. This point is now discussed in the manuscript (line 645-650)

  • All three tables were not correctly inserted into the text and were therefore not fully reviewable. Moreover, explanation of abbreviations are missing at the end of most table and graph captions (e.g. Table 1: what does “J” stand for?; Figure 3: “NI” = not infected?; etc.).

We apology for this problem that have occurred during formatting of the manuscript. The three tables have been re-formatted to be fully visible. “J” should be “D” that stands for “day”. We apologize for this this mistake. Table 1 has been corrected accordingly and abbreviations are now corrected and explained appropriately (line 181-183).

 Minor comments:

  • Line 13: HEV is only limited zoonotic, as HEV-1 and HEV-2 are restricted to humans (as later correctly stated by the authors in line 44-45).

The term “zoonotic” has been deleted (line 13).

  • Line 43-44: Not only four but also five HEV genotypes infect humans. The fifth is HEV-7 (PMID: 26551551)

The text has been modified to state more clearly that four main genotypes infect humans (line 49) and that a fifth HEV genotype, HEV-7, is also able to infect humans (line 60-61) with the appropriate reference (Lee et al, 2016).

  • Line 53: Only HEV-1 poses a severe threat to pregnant women.

The text now specifies that only HEV-1 is a severe threat to pregnant women (line 64).

  • Line 54: Chronification has also been reported for HEV-7.

Chronic infection in a patient infected with HEV-7 is now mentioned in the text (line 68-69).

  • Line 67-68: Ribavirin-induced HEV mutations do not confer resistance. The mutated virus is still sensitive to the drug (PMID: 27222534) and mutations do not affect viral clearance (PMID: 31793638).

As ribavirin can cause mutations within the HEV genome during treatment (Todt et al, 2016; PMID: 27222534), mutants that are more resistant to the drug could appear. However, we agree with Reviewer 1 that so far, the mutations detected have no impact on drug sensitivity or viral clearance. The text has been modified and appropriate references added in order to clarify this point (line 79-85).

  • Line 135-136: Were HEV positive cells passaged during the 382 days long maintenance or was the medium simply refreshed every 2-3 days?

Cells were not passaged during the 382 days and half of the medium refreshed every 2 to 3 days. This is now clearly stated in the text (line 166-167).

  • Line 139: I guess Table 1 should be in section “2.3” rather than “2.6” (line 181).

The table has been moved to section 2.3.

  • Line 170-171: Since the MGB modification of the probe was used as stated in line 176, Garson et al., 2012 (PMID: 22871672) should be cited in this context.

The appropriate reference (reference 34, Garson et al, 2012) has been added (line 211-212).

  • Line 207: All ibidid µ-plates are made of polymer and only µ-slides are made of glass.

We apologize for this mistake. The term “glass” has been deleted (line 247).

  • Line 226: Please add more details about the recombinant antigen (derived from which ORF? Length?)

The recombinant antigen used for the ELISA is proprietary. The manufacturer’s protocol only specifies that an antigen that is highly conserved between different HEV strains is used and refers to the ORF2.1 fragment (amino acids 394 to 660 of the capsid protein) (Anderson et al, 1999: PMID: 10488771; Chen et al, 2005; PMID: 15879020 and Hu et al, 2008; PMID: 18495846). These details and a reference have been added to the text (line 265-270).

  • Line 251: How long were the reads?

The mean read length was 79 nucleotides. This detail has been added to the material and method (line 295).

  • Line 289: Cholangiocytes, not choliangocytes.

This mistake has been corrected (line 341).

  • Line 374: 1x108 GE per what?

The text now specifies that pigs were inoculated orally with a total of 1x108 GE per animal in a volume of 10 ml (line 433-434).

  • Line 442-443: Why do the authors think, cDNA clones would replicate more efficiently than clinical isolates? Their data as well as data from Schemmerer et al., 2019 (PMID: 31141895) indicate that clinical isolates also replicate very efficiently.

It can sometimes be very difficult to grow clinical isolates (because of the low viral load, toxicity, availability of the inoculum) in cell culture. However, we agree that data from Schemmerer et al, 2019 and other studies, including ours, show that some clinical isolates can replicate very efficiently in vitro. This sentence has then been changed to better reflect this point (line 512-514).

  • Line 450-451: What was the exact viral load of the faecal suspension? In addition, HEV-3f instead of HEV-3 would be more informative to the reader.

The viral load of the fecal suspension was 1.4x109 GE/ml and not GE/g as written in the material and method of the original manuscript (section 2.2, line 149). This mistake has been corrected. HEV-3f is now used to describe the virus instead of HEV-3.

  • Line 489-492: If I recall that correct, these insertions were only (or mainly?) common in subtype 3f strains. Could the authors provide more information about that in this context?

Long insertions of 69 to 87 nucleotides were only found in HEV-3f isolates. Insertions of 39 and 18 nucleotides were also identified in HEV-3e and HEV-3a isolates, respectively. The text now provides more information about the subtypes concerned by these insertions (line 584-587).

  • Line 556-558: Before stating the usefulness for detecting infectious particles in samples, the authors should evaluate the sensitivity of this cell culture system. Did the authors titrate the HEV-3f strain and were other HEV strains also isolated?

Titrations of the HEV-3f strain were not performed (in focus forming unit). However, replication was detected using our cell culture system from an MOI of 0.1, which corresponds to the addition of ~3x105 GE par well containing ~3x106 differentiated HepaRG cells. As indicated above, we also infected successfully differentiated HepaRG cells with an HEV-1 isolate and field HEV-3f isolates (derived from the liver of an infected pig and a wild boar). Further experiments are currently in progress to evaluate the usefulness of our cell culture system to detect infectious particles in samples. These results will be presented in a future publication. This point is now clarified (line 660-663).

  • Figure 1: The authors state, they maintained the infection for up to 382 days but only show data for day 0-100. Please add the missing data.

Infection was maintained for up to 383 days for passage 2 and for 95 to 321 days for the other passages as indicated in table 1. To better visualize the logarithmic phase, we only showed the data from day 0 to 100 in the previous version of the manuscript. The whole data are now presented in an additional panel in Figure 1 (Figure 1b). We have also analyzed additional samples and consequently updated table 1 (HEV concentration at the latest time point is now shown) and figure 1b-c.

  • Figure 5a: The tree should be recalculated using the latest HEV reference set by Smith et al., 2020 (PMID: 32469300).

The tree has been recalculated using the latest reference set from Smith et al, 2020 and additional HEV-3 subtypes (Figure 5a). This reference is now added in the material and method (line 318-321).

  • Table 3: HEV strain 47832 is a subtype HEV-3c not HEV-3i. In addition, subtype information is missing for JE03-1706 and HE-JF5. Please add the correct subtype according to Smith et al., 2020 (PMID: 32469300).

The correct subtypes have been added according to Smith et al., 2020 for HEV strain 47832 (HEV-3c), JE03-1706 (HEV-3b) and HE-JF5 (HEV-4c) (Table 3).

  • Discussion: Are the ribavirin concentrations leading to viral clearance in this work similar to that of other publications?

In other publications, ribavirin concentrations of up to 200 μM were used and clearance was observed with concentration equal or superior to 100 μM depending on the in vitro model used. In Huh7.5 transfected with different HEV replicons, an inhibitory concentrations 90 (IC90) of 50 to 64 μM were determined after 42 hours treatment (Todt et al, 2018; PMCID: PMC7113770).  In infected PLC/PRF/5 cells, it was shown that HEV growth was suppressed by 160 μM ribavirin but not by concentrations equal or inferior to 80 μM even after 28 days of treatment (Nishiyama et al, 2019; PMID: 31004661). In Huh7 cells infected with the Kernow-C1 p6 virus, ribavirin inhibited HEV replication at 12 and 20 days post-infection in a dose-dependent manner. At a concentration of 100 μM, a 4-log10 reduction in viral titer (HEV GE/ml supernatant) was found without complete clearance of the virus (Debing et al, 2014; PMID: 24145541). In addition, HEV replication was inhibited by more than 95% after treatment with 100 μM ribavirin in human primary intestinal cells infected by different strains of HEV (Marion et al, 2019; PMID: 31727684). These findings are therefore in agreement with our data showing that a concentration of 10 μM leads to the reduction of HEV RNA concentration in the supernatant by more than 80% after 23 days of treatment and that clearance occurs with concentration equal or superior to 50 μM. This point is now detailed and discussed in the text (line 408-410 and line 560-573).

  • Titres represent dilutions but in this work, viral loads are indicated as GE/ml and therefore as concentrations.

“Titres” have been changed to “concentrations” in the whole manuscript.

Reviewer 2 Report

In this manuscript, Pellerin and colleagues report a useful the human HepaRG hepatic cell line in order to development persistent hepatitis E virus infection model. They showed that they could infect naive HepaRG cells and pigs after 6 passages in a HepaRG cell-based system, and that ribavirin decrease HEV in HEV-infected HepaRG cells.
Although the study have interesting aspect, I has some concerns regarding the interpretation of data.

My comments are listed below.

1. What does the "J" in Table 1 represent?

2. Why did you not unify the amount of virus inoculated into the cells for each passage in Table 1 and Figure 1b?

3. It's necessary to show error bars in Figure 2a.

4. In Fig. 1b and Fig. 2a, HepaRG cells are infected the same virus in the same moi, but why is the amount of virus detected in Fig. 2a so lower than Fig. 1b?

5. What is the IC50 value of ribavirin for HEV-infected HepaRG cells in Fig. 3?

6. The authors argue the 25 mutations shown in Table 2 may be neccesary for repeated passage of the virus increased infection efficiency in Fig. 1b. Is it possible to perform a full sequence analysis from passage 2 to passage 6 to show that mutations are accumulating as the passage progresses?

Author Response

Reviewer 2

In this manuscript, Pellerin and colleagues report a useful the human HepaRG hepatic cell line in order to development persistent hepatitis E virus infection model. They showed that they could infect naive HepaRG cells and pigs after 6 passages in a HepaRG cell-based system, and that ribavirin decrease HEV in HEV-infected HepaRG cells.
Although the study have interesting aspect, I has some concerns regarding the interpretation of data.

 We greatly thank Reviewer 2 for his/her comments and very useful suggestions. The manuscript has been modified to clarify all the points raised by reviewer 2. The modifications made to the manuscript are highlighted using the "track changes" feature in the word processing application. Responses to the reviewer comments are presented below in red color. The lines indicated refer to the revised manuscript including the “track changes”.

My comments are listed below.

  1. What does the "J" in Table 1 represent?

We apologize for this this mistake. “J” should be “D”, for “day”. Table 1 has been corrected accordingly (line 181-183).

  1. Why did you not unify the amount of virus inoculated into the cells for each passage in Table 1 and Figure 1b?

We usually performed the infection using fresh supernatant from a previous passage and determined the viral load of the inoculum afterwards. This led to differences in the amount of virus inoculated between the different passages.

  1. It's necessary to show error bars in Figure 2a.

Error bars have been added to Figure 2a and corrected in Figure 3 as standard error of mean was presented instead of standard deviation in the initial manuscript.

  1. In Fig. 1b and Fig. 2a, HepaRG cells are infected the same virus in the same moi, but why is the amount of virus detected in Fig. 2a so lower than Fig. 1b?

Data displayed in Fig. 1b et Fig. 2a were performed separately and variations can occur between experiments.

  1. What is the IC50 value of ribavirin for HEV-infected HepaRG cells in Fig. 3?

To calculate the IC50 value, a larger set of ribavirin concentrations need to be tested. Data from Figure 3 show that a concentration of 10 μM leads to the reduction of HEV RNA concentration in the supernatant by more than 80% after 23 days of treatment and that clearance occurs with concentration equal or superior to 50 μM. In other publications, ribavirin concentrations of up to 200 μM were used and clearance was observed with concentration equal or superior to 100 μM depending on the in vitro model used. In Huh7.5 transfected with different HEV replicons, an inhibitory concentrations 90 (IC90) of 50 to 64 μM were determined after 42 hours treatment (Todt et al, 2018; PMCID: PMC7113770).  In infected PLC/PRF/5 cells, it was shown that HEV growth was suppressed by 160 μM ribavirin but not by concentrations equal or inferior to 80 μM even after 28 days of treatment (Nishiyama et al, 2019; PMID: 31004661). In Huh7 cells infected with the Kernow-C1 p6 virus, ribavirin inhibited HEV replication at 12 and 20 days post-infection in a dose-dependent manner. At a concentration of 100 μM, a 4-log10 reduction in viral titer (HEV GE/ml supernatant) was found without complete clearance of the virus (Debing et al, 2014; PMID: 24145541). In addition, HEV replication was inhibited by more than 95% after treatment with 100 μM ribavirin in human primary intestinal cells infected by different strains of HEV (Marion et al, 2019; PMID: 31727684). These findings are therefore in agreement with the data presented in this manuscript. This point is now discussed in details in the text (line 560-573).

  1. The authors argue the 25 mutations shown in Table 2 may be necessary for repeated passage of the virus increased infection efficiency in Fig. 1b. Is it possible to perform a full sequence analysis from passage 2 to passage 6 to show that mutations are accumulating as the passage progresses?

In the discussion, we stated that one or several combinations of these 25 mutations might have improved infection efficiency (line 618-621). However, this hypothesis can only be tested using reverse genetic. Then, the impact of individual or combination of mutations on viral fitness could be determined in the backbone of the parental virus. Unfortunately, a reverse genetic system is not available yet with the FR-HuHEVF3f strain. This point is now further discussed (line 645-650).

Reviewer 3 Report

Title: Characterization of a cell culture system of persistent hepatitis E virus infection in the human HepaRG hepatic cell line

Summary: Pellerin et al presented a model for studying HEV infection biology using HepaRG cells. They used HEV-3 isolate (FR-HuHEVF3f) and cultured it in HepaRG cells and demonstrated higher titers at passage 7 which when tested in vivo in pigs was able to infect the pigs. They further demonstrated the sequence difference between the original isolate and the passage 7, thus suggesting 8 nonsynonymous mutations provide adaptability and higher infection ability to the virus. Furthermore, they demonstrated inhibitory effect of ribavirin on HEV replication onto HepaRG cells. The study seems interesting as it would be useful in studying other field isolates and identifying appropriate drugs against chronic HEV infection.

Minor revisions:

Line 43 – ORF2 codes for capsid protein and secreted form of capsid protein

Reference - Yin X, Ying D, Lhomme S, Tang Z, Walker CM, Xia N, Zheng Z, Feng Z. Origin, antigenicity, and function of a secreted form of ORF2 in hepatitis E virus infection. Proc Natl Acad Sci U S A. 2018 May 1;115(18):4773-4778. doi: 10.1073/pnas.1721345115. Epub 2018 Apr 18. PMID: 29669922; PMCID: PMC5939091.

Line 75 – please include the recent in vitro work and others too

Reference - Todt D, Friesland M, Moeller N, Praditya D, Kinast V, Brüggemann Y, Knegendorf L, Burkard T, Steinmann J, Burm R, Verhoye L, Wahid A, Meister TL, Engelmann M, Pfankuche VM, Puff C, Vondran FWR, Baumgärtner W, Meuleman P, Behrendt P, Steinmann E. Robust hepatitis E virus infection and transcriptional response in human hepatocytes. Proc Natl Acad Sci U S A. 2020 Jan 21;117(3):1731-1741. doi: 10.1073/pnas.1912307117. Epub 2020 Jan 2. PMID: 31896581; PMCID: PMC6983376.

Reference - Yadav KK, Boley PA, Fritts Z, Kenney SP. Ectopic Expression of Genotype 1 Hepatitis E Virus ORF4 Increases Genotype 3 HEV Viral Replication in Cell Culture. Viruses. 2021 Jan 7;13(1):E75. doi: 10.3390/v13010075. PMID: 33430442.

Reference - Knegendorf L, Drave SA, Dao Thi VL, Debing Y, Brown RJP, Vondran FWR, Resner K, Friesland M, Khera T, Engelmann M, Bremer B, Wedemeyer H, Behrendt P, Neyts J, Pietschmann T, Todt D, Steinmann E. Hepatitis E virus replication and interferon responses in human placental cells. Hepatol Commun. 2018 Jan 8;2(2):173-187. doi: 10.1002/hep4.1138. PMID: 29404525; PMCID: PMC5796324.

Reference - Marion O, Lhomme S, Nayrac M, Dubois M, Pucelle M, Requena M, Migueres M, Abravanel F, Peron JM, Carrere N, Suc B, Delobel P, Kamar N, Izopet J. Hepatitis E virus replication in human intestinal cells. Gut. 2020 May;69(5):901-910. doi: 10.1136/gutjnl-2019-319004. Epub 2019 Nov 14. PMID: 31727684.

Line 77-79 – Strong statement, the above-mentioned papers have shown good replication. Better to rewrite it reducing some intensity.

Line 90 “an isolate”

Line 138 – describe MOI, its been done later at line 291

Line 170 – better to be consistent with the referencing style.

Line 171 – is it necessary to include 15 and 20 reference here?

Line 214 – the dilution of secondary antibody used?

Line 223 should it read “Diagnostics” rather than “Di-agnostics”?

Line 248 – eluted

Line 255 is “bwa” an acronym that can be defined?

Line 373 – Virus obtained was from supernatant or cell lysates?

Line 384 – Fig. 4, better to include what NI means in figure legend

Line 432 -433 – have been repeated (its also in the introduction)

Line 444 – Hence, there is a

Line 449 – better to be consistent throughout the manuscript (use either biliary-like epithelial or cholangiocytes

Line 453-455 – it is already been stated in above in methodology / results

Line 462-469 – Why is there a difference between the previous and present study? Need to discuss it in detail!

Line 546 – better to be consistent in using HVR or PPR

More information on the starting material may be helpful.  Was the starting inoculum from a chronically infected patient? Is the output virus a predominant type or are significant output subspecies observed?

Can a genetically alterable laboratory strain of HEV such as Kernow-C1 progress to chronicity in this model?  Have the authors made any attempts to propagate strains other than the 3f patient derived samples?

The study would have been significantly more interesting if it would have utilized other strains.  So, variable nonsynonymous/synonymous rate ratios among lineages would indicate adaptive evolution. It is a good presentation of nonsynonymous mutations in various HEV-3 isolate in Table. 3. It would be better if variable nonsynonymous/synonymous rate ratios among different isolates would be mentioned thus it would be clearer that the several passage actually gave adaptive characteristics to FR-HuHEVF3f.

Author Response

Reviewer 3

Title: Characterization of a cell culture system of persistent hepatitis E virus infection in the human HepaRG hepatic cell line

Summary: Pellerin et al presented a model for studying HEV infection biology using HepaRG cells. They used HEV-3 isolate (FR-HuHEVF3f) and cultured it in HepaRG cells and demonstrated higher titers at passage 7 which when tested in vivo in pigs was able to infect the pigs. They further demonstrated the sequence difference between the original isolate and the passage 7, thus suggesting 8 nonsynonymous mutations provide adaptability and higher infection ability to the virus. Furthermore, they demonstrated inhibitory effect of ribavirin on HEV replication onto HepaRG cells. The study seems interesting as it would be useful in studying other field isolates and identifying appropriate drugs against chronic HEV infection.

 We greatly thank Reviewer 3 for his/her comments and very useful suggestions. The manuscript has been modified to clarify all the points raised by reviewer 3. The modifications made to the manuscript are highlighted using the "track changes" feature in the word processing application. Responses to the reviewer comments are presented below in red color. The lines indicated refer to the revised manuscript including the “track changes”.

Minor revisions:

Line 43 – ORF2 codes for capsid protein and secreted form of capsid protein

Reference - Yin X, Ying D, Lhomme S, Tang Z, Walker CM, Xia N, Zheng Z, Feng Z. Origin, antigenicity, and function of a secreted form of ORF2 in hepatitis E virus infection. Proc Natl Acad Sci U S A. 2018 May 1;115(18):4773-4778. doi: 10.1073/pnas.1721345115. Epub 2018 Apr 18. PMID: 29669922; PMCID: PMC5939091.

These details on ORF2 and the associated reference have been added to the text (line 44).

Line 75 – please include the recent in vitro work and others too

Reference - Todt D, Friesland M, Moeller N, Praditya D, Kinast V, Brüggemann Y, Knegendorf L, Burkard T, Steinmann J, Burm R, Verhoye L, Wahid A, Meister TL, Engelmann M, Pfankuche VM, Puff C, Vondran FWR, Baumgärtner W, Meuleman P, Behrendt P, Steinmann E. Robust hepatitis E virus infection and transcriptional response in human hepatocytes. Proc Natl Acad Sci U S A. 2020 Jan 21;117(3):1731-1741. doi: 10.1073/pnas.1912307117. Epub 2020 Jan 2. PMID: 31896581; PMCID: PMC6983376.

Reference - Yadav KK, Boley PA, Fritts Z, Kenney SP. Ectopic Expression of Genotype 1 Hepatitis E Virus ORF4 Increases Genotype 3 HEV Viral Replication in Cell Culture. Viruses. 2021 Jan 7;13(1):E75. doi: 10.3390/v13010075. PMID: 33430442.

Reference - Knegendorf L, Drave SA, Dao Thi VL, Debing Y, Brown RJP, Vondran FWR, Resner K, Friesland M, Khera T, Engelmann M, Bremer B, Wedemeyer H, Behrendt P, Neyts J, Pietschmann T, Todt D, Steinmann E. Hepatitis E virus replication and interferon responses in human placental cells. Hepatol Commun. 2018 Jan 8;2(2):173-187. doi: 10.1002/hep4.1138. PMID: 29404525; PMCID: PMC5796324.

Reference - Marion O, Lhomme S, Nayrac M, Dubois M, Pucelle M, Requena M, Migueres M, Abravanel F, Peron JM, Carrere N, Suc B, Delobel P, Kamar N, Izopet J. Hepatitis E virus replication in human intestinal cells. Gut. 2020 May;69(5):901-910. doi: 10.1136/gutjnl-2019-319004. Epub 2019 Nov 14. PMID: 31727684.

These studies are now referenced within the text alongside other more recent in vitro studies (reference 22-27, line 91-102).

Line 77-79 – Strong statement, the above-mentioned papers have shown good replication. Better to rewrite it reducing some intensity.

We have rewritten this sentence to moderate this statement and included details on recent studies showing good replication (line 96-98).

Line 90 “an isolate”

This mistake has been corrected (line 110).

Line 138 – describe MOI, its been done later at line 291

MOI (multiplicity of infection) is now described earlier in the text, when first used (line 109-110).

Line 170 – better to be consistent with the referencing style.

The referencing style has been modified to be consistent (line 205-206).

Line 171 – is it necessary to include 15 and 20 reference here?

We agree that these two references are not necessary here as a reference is already provided (reference 25) and details of the modifications made are given in the text. Reference 15 and 20 have then been removed.

Line 214 – the dilution of secondary antibody used?

The dilution of the secondary antibody used (1:5000) has been added (line 243).

Line 223 should it read “Diagnostics” rather than “Di-agnostics”?

It should be Diagnostics. This mistake has been corrected (line 263).

Line 248 – eluted

This mistake has been corrected (line 291).

Line 255 is “bwa” an acronym that can be defined?

“bwa” is an acronym for “Burrows-Wheeler Aligner” and is now defined in the text (line 299).

Line 373 – Virus obtained was from supernatant or cell lysates?

It is now specified in the text that the virus was obtained from the supernatant (line 431-432).

Line 384 – Fig. 4, better to include what NI means in figure legend

It is now included in the figure legend that NI means non-infected (line 451).

Line 432 -433 – have been repeated (its also in the introduction)

This sentence has been shortened to avoid repetition with the introduction (line 499-504).

Line 444 – Hence, there is a

This sentence has been modified accordingly (line 514).

Line 449 – better to be consistent throughout the manuscript (use either biliary-like epithelial or cholangiocytes

“Cholangiocyte” is now used throughout the manuscript (line 105, 341 and 520).

Line 453-455 – it is already been stated in above in methodology / results

This part of the conclusion has been synthesized to avoid repetition (line 522-527).

Line 462-469 – Why is there a difference between the previous and present study? Need to discuss it in detail!

In the previous study, a system using 3D Matrigel-embedded HepaRG cells that were infected with a HEV-3f isolate originating from an experimentally infected pig. Low HEV concentrations were detected extracellularly (1x103 GE/ml) and intracellularly (1x105 GE/ml) after 32 to 35 days of infection (Roger et al, 2012). In this study, we infected HepaRG cells differentiated via the addition of DMSO with a human HEV-3f isolate derived from the faces of a patient suffering from acute hepatitis E. Higher HEV concentrations were detected extracellularly (> 1x105 GE/ml) and intracellularly (1x106 GE/ug RNA) after 26 to 29 days of infection. In the present study, we also successfully maintained the infection for prolonged period and reached HEV RNA concentration in the supernatant of up to 1x109 GE/ml. Thus, the combination of DMSO-differentiated HepaRG and the human HEV-3f isolate allowed us to develop an efficient system of HEV replication and to passage the virus.  This is now discussed in more details in the discussion (line 541-549).

Line 546 – better to be consistent in using HVR or PPR

PPR has been replaced by HVR (line 642).

More information on the starting material may be helpful.  Was the starting inoculum from a chronically infected patient? Is the output virus a predominant type or are significant output subspecies observed?

As mentioned in the material and method (line 145-146), the starting inoculum was from a patient suffering from autochthonous acute hepatitis E. This detail is now corrected in the discussion (line 578). No further details are known about the origin of the inoculum and clinical history of the patient. The FR-HuHEVF3f isolate (GenBank accession number JN906974) has already been described and sequenced in a previous study from our group (Bouquet et al, 2012) as referenced in the text (reference 30, line 145). In this previous study, high-throughput sequencing was performed and detected the presence of 42 snp in the fecal sample from this patient, corresponding to 0.5% of the genome. The proportion of the HEV population displaying one particular SNP was lower than 20%. These details are now included within the text (line 153-158).

Can a genetically alterable laboratory strain of HEV such as Kernow-C1 progress to chronicity in this model?  Have the authors made any attempts to propagate strains other than the 3f patient derived samples?

We did not attempt to grow laboratory strain such as Kernow-C1 in our model. We have infected our cell model with other HEV isolates and have detected replication with a HEV-1 virus isolated from the feces of a patient suffering from acute hepatitis E and with other HEV-3f isolates derived from the liver of an infected pig and a wild boar. The concentrations of HEV RNA detected in the supernatant of cells infected with these isolates were lower than the ones obtained with the HEV-3f strain described in this study and passages of the virus was not attempted or not successful. The goal of this manuscript is to characterize the system of HEV replication that involved differentiated HepaRG and the FR-HuHEVF3f isolate as this cell/virus combination allows efficient replication. The isolation of additional strains will be the subject of a future publication. A sentence has been added in the text to mention this point (line 660-663).

The study would have been significantly more interesting if it would have utilized other strains.  So, variable nonsynonymous/synonymous rate ratios among lineages would indicate adaptive evolution. It is a good presentation of nonsynonymous mutations in various HEV-3 isolate in Table. 3. It would be better if variable nonsynonymous/synonymous rate ratios among different isolates would be mentioned thus it would be clearer that the several passage actually gave adaptive characteristics to FR-HuHEVF3f.

We agree with Reviewer 3 that it would be interesting to determine whether successive passages of the FR-HuHEVF3f isolate in HepaRG cells gave adaptive advantages to the virus. A nonsynonymous to synonymous nucleotide substitution rate ratio (dN/dS) of 0.24 was calculated (using http://services.cbu.uib.no/tools/kaks), suggestive of negative selective pressure. Nevertheless, this result accounts for the whole genome and it is possible that some specific sites were under positive selection. Studies have already reported that most of the HEV genome is under purifying selection (dN dS < 1) but that specific regions showed evidence of positive selection (dN/dS > 1): the hypervariable region (HVR) in ORF1 and the overlapping region of ORF2 (Brayne et al, 2017; PMID: 28202767, Purdy et al, 2012; PMID: 22545153). However, to fully study adaptive evolution of FR-HuHEVF3f and determine whether positive selection occurred at specific sites, additional experiments and analysis need to be performed with other isolates and larger sequence datasets. Moreover, to determine whether the non-synonymous mutations selected during successive passage have an impact on viral fitness, full-length infectious cDNA clones need to be constructed. Performing such experiments is out of scope of this study that is aiming at characterizing an efficient and relevant system of HEV replication that involves the combination of differentiated HepaRG cells and the FR-HuHEVF3f isolate. This point is now clarified in the manuscript (line and 645-650).

Round 2

Reviewer 1 Report

The authors addressed every comment to my entire satisfaction and significantly improved the manuscript. In particular, I want to thank the authors for shedding light on subtype specific lengths of insertions in the HVR and for discussing the different ribavirin concentrations in remarkable detail. Moreover, the added data to Figure 1 spanning the complete observation time is very interesting. All in all the manuscript now reads very well and certainly benefits to the field of HEV cell culture systems.